**METHOD**                                                                                    **Open Access**

# TT-Mars: structural variants assessment based on haplotype-resolved assemblies

Jianzhi Yang and Mark J.P. Chaisson[*] (ID)

*Correspondence:
mchaisso@usc.edu
Department of Quantitative and
Computational Biology, University
of Southern California, Los Angeles,
CA, USA

## Abstract

Variant benchmarking is often performed by comparing a test callset to a gold standard set of variants. In repetitive regions of the genome, it may be difficult to establish what is the truth for a call, for example, when different alignment scoring metrics provide equally supported but different variant calls on the same data. Here, we provide an alternative approach, TT-Mars, that takes advantage of the recent production of high-quality haplotype-resolved genome assemblies by providing false discovery rates for variant calls based on how well their call reflects the content of the assembly, rather than comparing calls themselves.

**Keywords:**  Structural variants, Benchmarking, Validation

## Background

Large-scale sequencing studies enable association of genetic variation such as common structural variations (SVs) [1] or rare variation of any [2] with traits and diseases. SVs include deletions, insertions, duplications, and rearrangements at least 50 bases that as a class have a considerable role in genetic diversity [3, 4], developmental disorders [5, 6], and cancer [7]. Compared to variations such as SNVs, SVs have been historically more difficult to detect using high-throughput short-read data, particularly due to the vast diversity of sizes and breakpoint complexity that SVs span [8]. In recent years, single-molecule sequencing (SMS) has been used to generate high-quality structural variant callsets [9–12] because long reads or their de novo assemblies span SVs, particularly in complex and repetitive regions [13, 14].

  Despite recent performance gains in SMS instrument throughput [15, 16], short-read sequencing remains the primary technology for sequencing large populations [1, 17]. Many algorithms have been developed to detect SVs using different types of information from short-read sequencing [18–20], but there are reports of both high rates of false positives [13] and low recall rates [9]. Additionally, detecting complex SVs is still challenging [9], and callers can produce different SV callsets on the same genome sample [21], which makes it more difficult to output a complete high-quality SV callset.

The limitations and challenges in SV detection, combined with an importance in clinical sequencing, motivates the need for efficient and precise tools to evaluate SV callset accuracy. A common framework for this benchmark is to establish a ground truth as a consensus between calls from multiple sequencing technologies and SV discovery algorithms and to compare new calls to the ground truth. One of the first approaches to generate a gold standard was the svclassify method that used machine learning to classify SVs as true positive (TP), false positive (FP), or unclear [22]. Recently, the Genome in a Bottle Consortium (GIAB) made a high-quality benchmark set of large ($\geq$ 50 bp) insertions and deletions using multiple SV callsets produced by a wide range of analysis methods [21]. This benchmark set can be used to evaluate arbitrary SV callsets generated by different combinations of algorithms and sequencing data. The accompanying method, truvari [21], is used to compare SV calls based on agreement between the breakpoints in the test and benchmark calls, with the option to compare sequences of variants. To be considered a true positive, the test variant must be within a specified size and distance of a ground truth call [21]. This approach has been an invaluable standard for benchmarking method accuracy; however, the notion of comparing callsets may be considered as a proxy comparison against the overarching goal of determining how well an algorithm estimates the content of a genome with a particular sequence input. In repetitive regions, there may be multiple placements of breakpoints that have equal support for a variant or similar placement of breakpoints that depend on the parameters for scoring alignments [23]. In part due to the breakpoint degeneracy, and difficulty in calling variants in repetitive regions, the benchmark callset excludes many variants in repetitive regions [21], although repetitive regions are enriched for SVs [10].

As de novo assembly quality has increased with improvements in algorithms and SMS technologies, assemblies have been used to generate SV callsets [24, 25]. SV calls based on haplotype-resolved assemblies are factored into the GIAB truth set, and haplotype-resolved assemblies have been used to create benchmark callsets for structurally divergent regions [24]; however, the difficulties in comparing calls in repetitive regions remain. Here, we propose an alternative approach to validate variant calls by comparing the sequences implied by an SV call to assemblies rather than comparing variant calls. This complements the validation method by the curated gold standard callset from the GIAB such that any genome with a haplotye-resolved assembly of sufficiently high quality may be included as a benchmark and can help validate SVs in repetitive regions because explicit breakpoints are not compared. While this resource has historically not been available, recent advances in single-molecule sequencing and assembly have enabled more routine generation of high-quality haplotype-resolved assemblies. The Human Genome Structural Variation Consortium (HGSVC) has generated 32 haplotype-resolved assemblies of human genomes at a Phred quality scale over 40 with contig N50 values over 25 Mb [12]. The Human Pangenome Reference Consortium is using a combination of Pacific Biosciences HiFi sequencing reads and the hifiasm method [26] to generate haplotype-resolved assemblies with base quality approaching QV50 and assembly N50 over 40 Mb. We have implemented our approach as a method, TT-Mars (structural variants assessment based on haplotype-resolved assemblies), which assesses candidate SV calls by comparing the putative content of a genome given an SV call against a corresponding haplotype-resolved de novo assembly.

We demonstrate that within the regions annotated as high-confidence by GIAB, TT-Mars has consistent results with the truvari analysis using the GIAB gold standard callset. TT-Mars also has consistent results with two other benchmarking methods: VaPoR [27], which is a long-read validation tool, and dipcall+truvari [28] using assembly-based variant calls as a gold standard. We demonstrate that compared to VaPoR, TT-Mars requires smaller input and shorter runtime to achieve comparable results, and validation of calls using TT-Mars is less dependent on alignment gap parameters compared to dipcall+truvari. Using 10 assemblies, we evaluate the distribution of call accuracy for three different short-read SV calling algorithms, LUMPY [18], Wham [29], and DELLY [19], as well as one long-read SV detection algorithm, pbsv [30]. The software is available at https://github.com/ChaissonLab/TT-Mars.git, which provides a utility to download assemblies and alignment maps.

## Results

### Validation of calls on GIAB HG002 assembly

On human genome sample HG002, we used TT-Mars to validate insertion, deletion, inversion, and duplication calls produced by four methods: LUMPY, Wham, DELLY, and pbsv (Table 1). Translocation/break-end calls and calls on the Y-chromosome were ignored. The runtime scales by the number of calls, most short read data sets can be evaluated within 1 h using a single core, and pbsv (long-read) callsets within 9 h, both using up to 28 G of memory.

The analyzed rates are in the range of 92.2–96.7%. Among the analyzed calls, the TP rates range between 78.4 and 90.0%. Among the calls that are not analyzed (NA), a majority of calls (65.7–78.4%) are in regions which are not covered by one or both assemblies and 14.8–31.4% are in centromeres. Figure 1a shows an example of scatter plot of validation results from TT-Mars, where the TP and FP calls are separated as two clear clusters. Figure 1c gives the distribution of length of calls that are analyzed and missed by TT-Mars. The distribution conforms with the length distribution of the benchmark SV callset [21].

We compared the TT-Mars results to the GIAB HG002 benchmark callset validated using truvari (Table 2) set to search for calls within 1kb (e.g., `refdist`) and not using sequence identity comparison. For long-read callsets, truvari was used with the option that compares the sequences of SV for validation. Since the GIAB benchmark set only contains deletions and insertions, validation of other types of SVs are not included. Generally, the two validation methods have consistent TP and FP results. For SV that are evaluated by both methods (e.g., excluding the NA calls), 96.0–99.6% of calls have the same classification results in the four callsets (Fig. 1b). For the three short read callsets, there is a net increase of 121–1476 calls analyzed by TT-Mars, and on the pbsv callset,

**Table 1** TT-Mars results on callsets from four SV discovery algorithms on HG002, including all calls larger than 10 bp produced by each method

| Callers | TP | FP | NA | Total |
|---|---|---|---|---|
| Wham | 1130 | 144 | 74 | 1348 |
| LUMPY | 2990 | 825 | 324 | 4139 |
| DELLY | 12,784 | 1833 | 812 | 15,429 |
| pbsv | 45,625 | 5057 | 1741 | 52,423 |

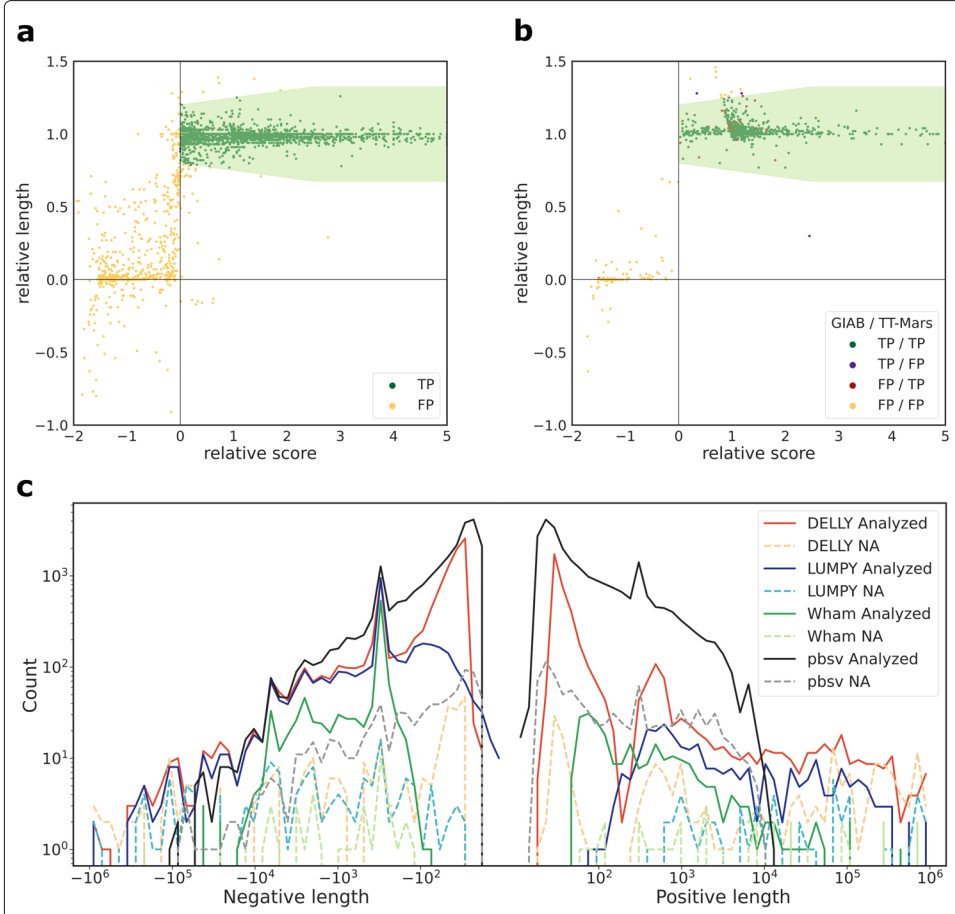

**Fig. 1** An example of classification of individual deletion calls by TT-Mars on HG002 made by DELLY (**a**) and LUMPY (**b**). Each dot represents a candidate call. The classification boundaries determined empirically for human data are shown by the light green region. **b** Different classifications made by TT-Mars and GIAB+truvari. **c** The length distribution of SVs in the HG002 callsets by four SV detection algorithms. The distribution of calls that are analyzed or where TT-Mars does not provide an annotation (NA) are shown

**Table 2** Comparison of TT-Mars and GIAB benchmarks on HG002. The two methods have the same classification results on more than 96% of the analyzed calls, while TT-Mars analyzed more candidate calls on all the four callsets. A length filter (50 bp to 10 Mbp) is applied, and only deletions and insertions are included to match the truvari parameter settings

| | | TT-Mars | | | | | | | |
|---|---|---|---|---|---|---|---|---|---|
| | | Wham | | | | LUMPY | | | |
| | | TP | FP | NA | Sum | TP | FP | NA | Sum |
| | TP | 882 | 3 | 21 | 906 | 2091 | 8 | 51 | 2150 |
| GIAB | FP | 1 | 13 | 0 | 14 | 22 | 218 | 3 | 243 |
| | NA | 118 | 24 | 20 | 162 | 582 | 215 | 99 | 896 |
| | Sum | 1001 | 40 | 41 | 1082 | 2695 | 441 | 153 | 3289 |
| | | DELLY | | | | pbsv | | | |
| | | TP | FP | NA | Sum | TP | FP | NA | Sum |
| | TP | 2927 | 11 | 57 | 2995 | 8903 | 94 | 191 | 9188 |
| GIAB | FP | 4 | 412 | 6 | 422 | 286 | 165 | 16 | 467 |
| | NA | 1179 | 360 | 145 | 1684 | 9710 | 1463 | 721 | 11,894 |
| | Sum | 4110 | 783 | 208 | 5101 | 18,899 | 1722 | 928 | 21,549 |

TT-Mars analyzes 10,966 more calls (Additional file 1: Fig. S1 and S2). The regions annotated as confident for validating variant calls by TT-Mars and excluded by truvari include 141 Mbp. These sequences predominantly overlap segmental duplications (99 Mbp) and highly repetitive sequences such as variable-number tandem repeats (4 Mbp).

**Performance on 10 HGSVC sample genomes**

Gold-standard callsets such as those produced by the GIAB require intense effort to create. Because TT-Mars only needs haplotype-resolved assemblies, multiple samples may be evaluated in order to give benchmark results as a distribution. Recently, the HGSVC has generated 32 high-quality haplotype-resolved assemblies of human genomes [12]. We used TT-mars to evaluate three short-read and one long-read SV-discovery algorithms on ten samples sequenced and assembled by the HGSVC (Fig. 2a). The pbsv callsets, except the HG002 callset, were generated by the HGSVC [12]. Broadly, these calls are generated using the pbmm2 alignment method (using minimap2 [31]) and an SV detection algorithm designed for PacBio raw read data. A second long-read SVs discovery method, cuteSV, had a low TP rate across every evaluation method, and is excluded from the results. The number of analyzed, true positive, and genotype matched sites assessed by TT-Mars are consistent across different samples, for both the short and long read callsets. The fraction of genome that is excluded due to coverage of assemblies is from 9.3 to 10.2% (Additional file 1: Table S1). The fraction of SV calls that are analyzed counts calls that are made by an algorithm in less repetitive regions of the genome that are amenable to long-read assembly, in particular, excluding centromeres and high-identity long segmental duplications. Wham callsets have the highest analyzed rate (93.3% on average) among the three short reads algorithms. The pbsv callsets have the analyzed rate in the range of 93.0–95.2%. The TP rates on short-read callsets range between 63.9 and 85.2%, while pbsv calls have TP rates from 71.4 to 94.3% considering all calls, and 91.7–94.8% for calls ≥ 30 bases. The sample HG00513 has lower TP rates among all short read SV discovery algorithms, but also has roughly double the input size of other samples that may have required parameter optimization (Additional file 1: Figs. S3 and S4). DELLY and pbsv output geno-

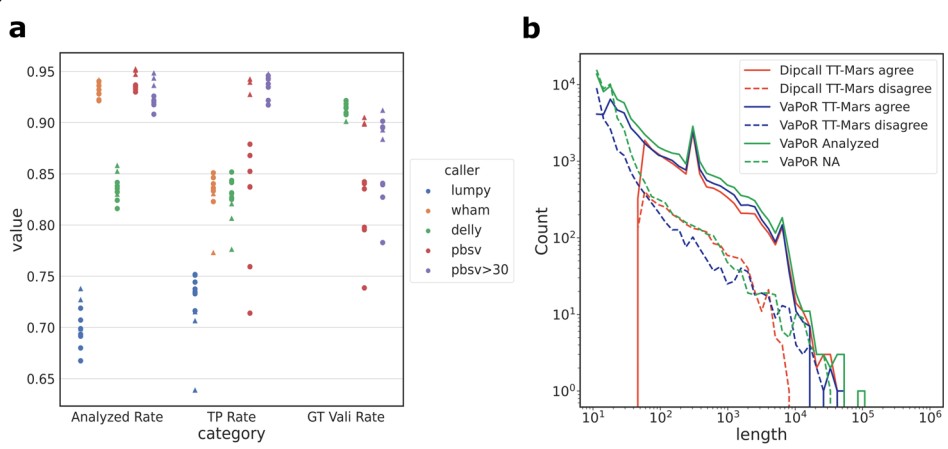

**Fig. 2  a**, SV metrics for four algorithms on 10 HGSVC genome samples, triangles mark samples using HiFi assemblies. Benchmark results are given as a distribution by TT-Mars. **b** The length distribution of the results by dipcall+truvari, TT-Mars, and VaPoR for pbsv HG00096 calls. The red solid and dashed lines indicate that dipcall+truvari and TT-Mars congruent calls have similar length distributions. The green and blue dashed lines show that VaPoR has more NA and disagreed results with TT-Mars for small SVs (size < 100 bases)

type. Among the TP calls, genotype matched rates of DELLY are from 90.1 to 92.1% and of pbsv are from 73.9 to 90.5% (78.3–91.2% for SV $\geq$ 30bp). TT-Mars also reports the following recall results: The average recall of Wham, LUMPY, DELLY, and pbsv are 7.7%, 13.1%, 16.0%, and 88.7%, respectively.

The availability of assembled sequences allows TT-Mars to validate duplications, specifically checking the genomic organization is in tandem or interspersed. The average number of duplications called by Wham, LUMPY, DELLY, and pbsv on the ten HGSVC genome samples are 1463, 4577, 761, and 1458, respectively. Duplications that are tandemly organized may be validated as inserted elements at the location of the insertion because of the tail-to-head organization. Interspersed duplications are more challenging to validate because the target destination is unknown. Because the GIAB benchmark callset does not include duplication calls, we estimated the performance of TT-Mars using simulations. We ran 1000 simulations of interspersed duplications with an exact sequence copy and length uniformly taken from 100 to 100,000 on the HG002 assembly. Not all duplications were analyzed by TT-Mars because there were no requirements placed on the target site of duplications. In total, 82.8% duplications were analyzed, of which 98.9% were validated (Table 3), indicating that interspersed duplications from non-repetitive euchromatic regions of the genome may be validated by TT-Mars. We also simulated 1000 false duplications; TT-Mars analyzed 88.2% of them of which all were annotated as FP (Table 3).

### Comparison to VaPoR and dipcall

We compared our method to two other approaches for validating variant calls that use information from long reads or their assembly: VaPoR, which compares SV calls with individual long reads, and dipcall+truvari, where a callset from a de novo assembly is used as a gold standard for evaluating an SV callset. Both methods are compared with TT-Mars on callsets produced by the four SV discovery methods (Wham, LUMPY, DELLY, and pbsv). Only insertion and deletion variants were in dipcall output, so that other types are not included in the comparison.

Due to the runtime of VaPoR, we limited analysis of data from one sample with each SV discovery method (Table 4). On short read datasets, TT-Mars and VaPoR provide the same validation result on 80.5–92.0% for calls analyzed by both methods; however, there was a net increase of 302–1863 calls that could be analyzed by TT-Mars. On the long read dataset, the two methods agree on 67.6% of calls analyzed by both methods, and TT-Mars analyzes 39,187 more calls. Because of the efficiency of dipcall, we were able to compare all SV callsets on the ten sample genomes. The validation of short-read callsets is similar, with the methods agreeing on validation results for 96.1–99.8% of calls analyzed by both methods (Fig. 3a). This is an expected result because of the relatively low repetitive nature of sequence where short-read algorithms detect SVs [23], and because both validation methods rely on the same assemblies. Similar to the VaPoR comparison, there was a net increase of 16–373 calls for which TT-Mars provides a classification compared to dipcall+truvari. On the ten pbsv callsets, TT-Mars and dipcall+truvari have same classi-

**Table 3** TT-Mars validation results of simulated true and false duplications

|                     | TP  | FP  | NA  | Total |
|---------------------|-----|-----|-----|-------|
| True DUP simulation | 819 | 9   | 172 | 1000  |
| False DUP simulation| 0   | 882 | 118 | 1000  |

**Table 4** Comparison of TT-Mars and VaPoR. The two methods agree on most calls and can analyze a similar number of calls across short-read callsets. On the long read callset, TT-Mars evaluates 39,187 additional variants, the majority of which are under 100 bases

| | | TT-Mars | | | | | | | |
| --- | --- | --- | --- | --- | --- | --- | --- | --- | --- |
| | | Wham | | | | LUMPY | | | |
| | | TP | FP | NA | Sum | TP | FP | NA | Sum |
| VaPoR | TP | 1503 | 98 | 76 | 1677 | 2614 | 288 | 282 | 3184 |
| | FP | 42 | 108 | 29 | 179 | 163 | 434 | 276 | 873 |
| | NA | 267 | 149 | 80 | 496 | 345 | 515 | 1370 | 2230 |
| | Sum | 1812 | 355 | 185 | 2352 | 3122 | 1237 | 1928 | 6287 |
| | | HG00171 | | | | HG00096 | | | |
| | | DELLY | | | | pbsv | | | |
| | | TP | FP | NA | Sum | TP | FP | NA | Sum |
| VaPoR | TP | 5211 | 405 | 480 | 6096 | 42,268 | 8694 | 3531 | 54,493 |
| | FP | 1009 | 627 | 581 | 2217 | 12,243 | 1320 | 870 | 14,433 |
| | NA | 2343 | 581 | 991 | 3915 | 37,646 | 5942 | 2765 | 46,353 |
| | Sum | 8563 | 1613 | 2052 | 12,228 | 92,157 | 15,956 | 7166 | 115,279 |
| | | HG03009 | | | | HG00096 | | | |

fication results for 83.8–86.9% of the analyzed calls (Fig. 3b), although TT-Mars analyzes 1497–2229 more insertions/deletions than dipcall+truvari.

The increased number of calls from long-read callsets provide granularity for which the validation approaches may be compared. We analyzed the classification results of TT-Mars, VaPoR, and dipcall+truvari on the pbsv callset on HG00096 (arbitrarily selected from the ten benchmark samples due to VaPoR runtime). There was no association between length (Fig. 2b) nor genomic organization (Additional file 1: Fig. S5) for whether TT-Mars and dipcall+truvari provided a validation; however, there is considerable enrichment in disagreement between methods in tandem duplications. For example, in HG00096, the 92.8% of calls with disagreeing validation overlap tandem repeats. The majority of the variants where VaPoR does not provide a call while TT-Mars does are less than 100 bases. To gauge the true results when two methods disagree, we randomly selected 50 (out of 2434) calls with TT-Mars TP and dipcall+Truvai FP results for manual inspection using IGV [32]. There are various sources for the discrepancy of validation

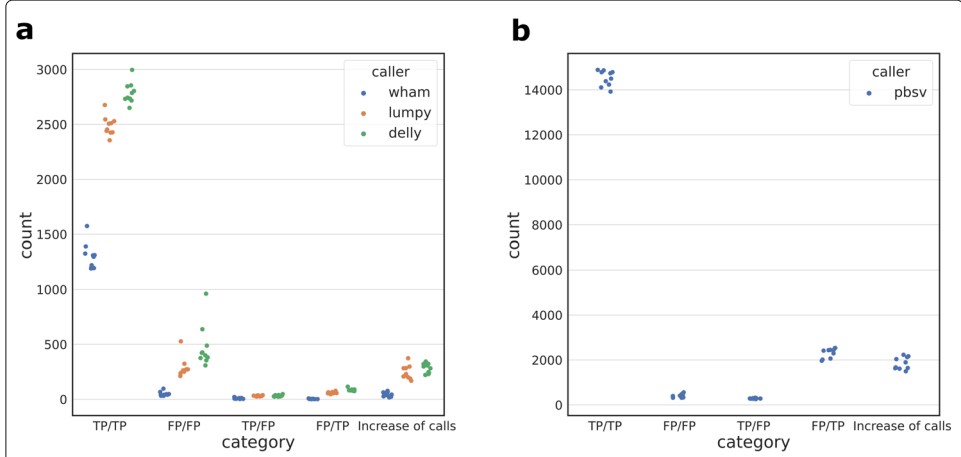

**Fig. 3** Comparison of dipcall+truvari and TT-Mars on short-read (**a**) and long-read (**b**) callsets of 10 sample genomes. The combinations of TP and FP for dipcall+truvari/TT-Mars annotations are given with a horizontal scatter to distinguish points in individual categories

result (Additional file 1: Fig. S6-S55). In 27 cases, a call made by pbsv is fragmented into multiple calls by dipcall (Fig. 4), highlighting how the assembly-based approach can account for validation in repetitive DNA where breakpoints are uncertain. In 19 other cases, there are correct variants nearby, and the validation depends on the genomic region considered by truvari. The remaining four cases do not show a matched SV in the alignments generated by minimap2 that are used for validation but do show matched SV(s) in the lra [33] based alignments, indicating how call-based validation rather than assembly-based validation is reliant upon agreement of gap penalty between detection and validation algorithms. We also manually inspected the converse case 50 (out of 280) calls with TT-Mars FP and dipcall+truvai TP results. Of these, 48 calls have matched SVs in the assembly alignment used by dipcall (using minimap2), and 45 calls do not have matched SVs in the assembly used by TT-Mars (lra). These represent the cases the assembly has unexpected complicated alignment to the reference leading to spurious gaps (Additional file 1: Fig. S56-S105).

## Discussion

Our method relies on highly accurate genome assemblies that do not have rearrangements that could miss a validation. We measured assembly by mapping all reads back to each haplotype and checking for contiguity of reads aligned across each position in the genome. Misassemblies not supported by reads are expected to show punctate drops

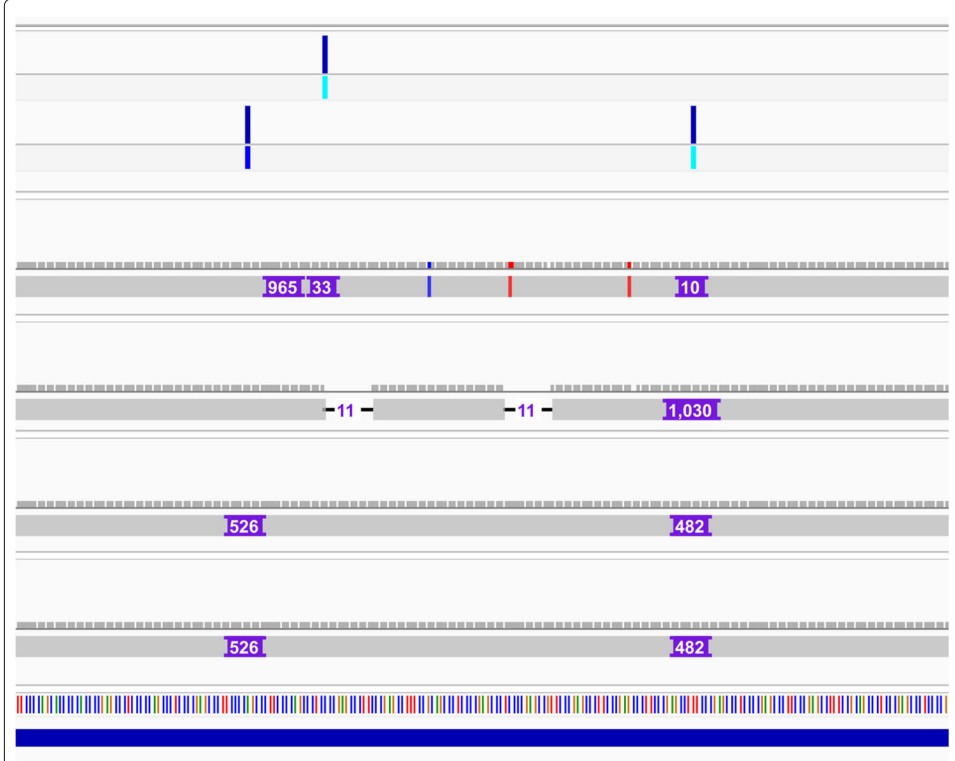

**Fig. 4** An example shows TT-Mars correctly validates an insertion but dipcall+truvari does not in a tandem repeat region, where the SV is split into two smaller insertions on the assembly alignment. The insertion with length 997 bases is on chromosome 4, coordinate 1046348. The first track is the pbsv callset, and the second track is the dipcall callset, followed by alignments used by TT-Mars (two tracks) and dipcall (two tracks). The bottom track is a tandem duplication region

in coverage. Overall, the assembly quality was quite high: only ten sites in HG002 were identified as possible misassemblies, and no SV overlapped these sites.

The majority of calls that are evaluated by both the GIAB + truvari and TT-Mars (96.0–99.6%) have the same classification, while evaluating 121–10,966 additional calls per dataset. Furthermore, we are able to evaluate 302–39,187 additional calls per callset compared to the long-read validation tool VaPoR, while not requiring users to download entire read data sets. We show that the assembly-content approach improves over benchmark data sets created from assembly alignment in repetitive regions of the genome because variants split into multiple calls from the assembly alignment are matched in the breakpoint searching procedure.

Duplication calls are a particularly challenging class of variant to validate. This has been done using comparison to array data [34], and more recently through semi-automated and automated visual inspection of calls [35]. Ground truth SV callsets created from assemblies detect duplications as insertions relative to the reference and are annotated by the site of insertion in the genome, compared to duplication calls in short-read callsets that annotate the source interval on the genome that is duplicated. By checking for both tandem duplications, which can be confirmed as insertions in assembly callsets, as well as interspersed duplications using the assemblies, TT-Mars can help provide insight on the classes of variation detected by SV discovery algorithms. One caveat of this analysis is duplications are enriched in repetitive sequences [36], and some validations will require high quality assemblies using ultra-long and accurate (HiFi) sequencing technologies.

Our analysis enables SV benchmarks to be reported as a spread over multiple genomes, rather than a point estimate on one curated data set, and provides insight on how extensible methods are on additional genomes from a single curated callset. Fortunately, most methods had accuracy within 5–10% across all genomes, indicating results are extensible across genomes and, for example, the results in current large-scale short-read based genomics studies likely hold valid. Currently, TT-Mars only validates deletion, insertion, inversion, and duplication calls. As additional methods emerge to detect more complex forms of variation [37], these models can be incorporated into the edit operations used by TT-Mars.

TT-Mars inherently provides a rough estimate of sensitivity because it does not fit into the paradigm of comparing inferred content, and requires variants to be called. This estimate simply considers false negatives as variants detected by haplotype-resolved assemblies that are not within the vicinity of the validated calls, and one should consider a class of variant that may have multiple representations when reporting results. Furthermore, because of the difference in scale of the number of variants discovered in short and long-read studies, it is important to consider the sensitivity of medically relevant variants compared to the total variant count.

## Conclusions

Here, we demonstrated an approach to validate SV using haplotype-resolved de novo assemblies where a call is validated by comparing the inferred content of a genome to the assembly, rather than comparing variant calls. This approach is implemented in the software TT-Mars, which is distributed along with accompanying data files required to benchmark variant calls on up to ten samples. It can validate SVs from both short- and long-reads calling algorithms with high accuracy compared to other methods, as well as

maintain a broad usability. TT-Mars enables SV benchmarks to be reported as a distribution over different genomes and provides information on the performance of methods on multiple samples without curated benchmark sets.

## Methods

### Workflow

TT-Mars uses a haplotype-resolved long-read assembly, and a lift-over map to the human reference to assess each candidate SV call (Fig. 5a–c). Each SV is considered independently as an edit operation on the reference genome (e.g., deletion, duplication) (Additional file 1: Fig. S106), and SV calls are evaluated by comparing both the reference modified by the edit operation and the original reference genome to the assembly (Fig. 5d–g).

To speed up the reference/assembly comparison, only the local region surrounding the SV call are compared using an assembly-genome orthology map constructed from whole-genome alignments (Fig. 5b). Each classification is made as a comparison between the relative scores of the alignment of the reference/modified reference and assembly at the SV locus, and/or the relative length, as detailed below.

### Validation

The haplotype-resolved assemblies are pre-processed by TT-Mars to generate an orthology map between the assembly and the reference for regions that are annotated as assembled without error and have a clear 1–1 relationship. The contigs are aligned using lra [33] with the options -CONTIG -p s, from which an orthology map at fixed intervals (20 bp by default) is generated using samLiftover (https://github.com/mchaisso/mcutils) (Fig. 5b). When two contigs have overlapping alignments on the reference, the alignment

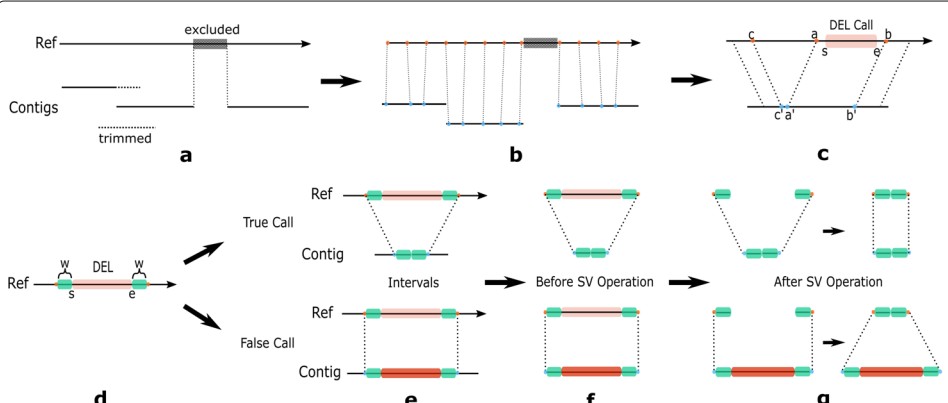

**Fig. 5** TT-Mars Workflow. **a** Assembly contigs are aligned to the reference, and the shorter of two overlapping contigs are trimmed to generate a unique mapping. Regions on the reference that are not covered by contigs are excluded. **b** The alignment is used to construct an orthology map at specific intervals (e.g., every 20 bases). **c** For an SV, called at an interval [*s*, *e*] on the reference, TT-Mars searches the orthology map for matches outside the interval that most closely reflects the length of the SV. In this example, the interval [*a*, *b*] immediately flanking a deletion SV maps to an interval on the assembly that does not reflect the SV, but a wider search in the orthology map shows that [*c*, *b*] spans a deletion in the assembly. **d–g** Validation details by a deletion example. **d** TT-Mars takes the candidate call with *w* flanking bases on both sides. The interval on the reference is compared with an interval on the assembly (**e**) before and after the SV operation. In **f** and **g**, the deletion operation removes the corresponding sequence on the reference and is validated if the modified reference is more similar to the assembly than the original

of the shorter contig is trimmed to exclude the overlap (Fig. 5a). Alignments of reads back to assemblies are used to flag potential misassemblies that should not be used to validate calls. Correctly assembled short intervals (e.g., 100 base windows) should have reads that map contiguously from at least one haplotype. We define an interval as correctly assembled if at least five reads are aligned starting/ending at least 1k bases before/after the interval. Intervals are misassembled if this condition does not hold, and the intervals in the flanking 1–3 kb are supported. These parameters are defined by the data to stringently detect misjoins in assemblies and not be affected by read mapping artifacts in repetitive DNA. Next, a high-confidence filter is created by excluding centromeres, intervals not mapped by both haplotypes in autosomal and X chromosomes in females, regions, and low-quality assembled regions flagged by raw-read alignments (Additional file 1: Fig. S107). SV calls that are flagged as analyzed if they are within the high-confidence filter.

The orthology map is accessed as a function Lookup($c, p$) that for a position $p$ on chromosome $c$ returns the assembly contig and position on the contig corresponding to the last position on the orthology map at or before $p$. To evaluate each SV call, we compare the local region surrounding the SV in either the original or modified reference to both assembled haplotypes (Fig. 5b, c). For calls with a defined starting position $s$ and ending position $e$, such as deletion calls (Fig. 5d), this region may be defined as a region on the reference defined by these positions, expanded by a small number of bases on either side ($w$ bases, $w = 500$), and the corresponding intervals on the assembly defined as Lookup($c, s - w$) and Lookup($c, e + w$). We assume that true-positive calls will produce a modified reference sequence that matches one or both haplotypes with high identity, as shown in the top figures in Fig. 5e–g, and that false-positive calls will produce a modified reference sequence that is different from both haplotypes as shown in the bottom figures in Fig. 5e–g. However, in highly repetitive regions, such as variable-number tandem-repeats, the alignment used to generate the orthology map may be different from that which is used to generate the SV call, and the interval defined directly from fixed offsets from the SV boundaries may not reflect this.

To account for potentially inconsistent breakpoints, we search for a combination of boundaries that maximizes the congruence between the modified reference and assembly. Two criteria are used to quantify the discrepancy of the reference and contig intervals. First, relative length is defined as:

$$\text{relative length} = \frac{\text{contig interval length} - \text{reference interval length}}{\text{sv length}}. \tag{1}$$

Second, the contig interval is aligned to the reference interval before and after the SV operation. The relative score is calculated by

$$\text{relative score} = \frac{\text{alignment score after} - \text{alignment score before}}{|\text{alignment score before}|}, \tag{2}$$

where alignment score after is the alignment score of the contig interval and the reference interval after the SV operation, and alignment score before is the alignment score before the SV operation. Figure 5c illustrates how inconsistent alignments in repetitive regions may cause incorrect inference of SV breakpoints in a lifted region.

TT-Mars validates separate classes of SVs by slightly different strategies. Deletion and sequence-resolved insertion calls are annotated as true positive if the relative length is

close to one and the relative score is positive. Specifically, the true positive region of these two types is defined as (Fig. 1a):

$$
\begin{cases}
-\alpha \times \text{relative score} + 1 - \beta \leq \text{relative length} \leq \alpha \times \text{relative score} + 1 + \beta, \\
\qquad\qquad\qquad\qquad\qquad\qquad\qquad\qquad 0 \leq \text{relative score} \leq \gamma; \\
1 - \delta \leq \text{relative length} \leq 1 + \delta, \qquad\qquad\qquad \text{relative score} > \gamma,
\end{cases} \quad (3)
$$

where $\alpha$, $\beta$, $\gamma$, and $\delta$ are empirical parameters.

Insertion SV calls that do not include the inserted sequence, such as many insertion calls made by short-read callers, are evaluated by relative length only, while inversion calls that do not change the length of the sequence are evaluated by relative score.

TT-Mars validates duplication calls by considering possibilities of both tandem duplication and interspersed duplication. A tandem duplication operation adds a copy of the duplicated sequence to the reference at the duplication locus in a tail to head orientation. The start and end coordinates of the duplication that are lifted over include both copies of the tandem duplication in true-positive calls. Similar to deletions and sequence-resolved insertions, it is validated by using both the relative length and the relative score. Interspersed duplications are evaluated by aligning the duplicated sequence to the assemblies using minimap2 [31] allowing for multiple alignments, and are validated if at least 90% of the duplicated sequence aligns to a sequence in the assembly annotated as an insertion relative to the reference.

The approach that compares genome content rather than callsets inherently does not support measurements of missed calls, or false negatives (FN). To enable reporting FN, TT-Mars provides the option by comparing against a callset offered as ground truth. Variants at least 50 bases produced by dipcall on autosomal and X chromosomes are used as the truth set for annotating false negatives. Calls overlapping with contigs that are trimmed out (Fig. 5a) are ignored since they may produce duplicated truth calls. Because variants from the candidate callset are not validated by matching calls, dipcall variants within a specified number of bases of validated candidate calls (default 1 kb) are considered matched. Remaining unmatched calls are reported as FN.

## Supplementary Information

---

**Additional file 1:** Supplementary figures and analysis.

**Additional file 2:** Review history.

---

### Acknowledgements
We would like to thank anonymous reviewers for their contribution to the peer review of this work.

### Peer review information

### Review history
The review history is available as Additional file 2.

### Authors' contributions
J.Y. and M.J.P.C. conceived the method. J.Y. implemented the software. J.Y. and M.J.P.C. wrote the manuscript. The authors read and approved the final manuscript.

### Funding
J.Y. is supported by U24HG007497, M.J.P.C. is supported by U24HG007497 and R01HG011649.

**Availability of data and materials**

The software is available at GitHub [38], which is licensed under the 3-Clause BSD License. The necessary lift-over files required to run TT-Mars as well as links to the original assemblies are provided through this repository. The source code is also available at Zenodo [39], with both the version used in the manuscript and the latest. Centromere and tandem repeat annotations were downloaded from the UCSC Genome Browser for GRCh38 from the cytoBandIdeo and simpleRepeats tables.

# Declarations

**Ethics approval and consent to participate**

Not applicable.

**Consent for publication**

Not applicable.

**Competing interests**

The authors declare that they have no competing interests.

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

## 
