## [**Additional file 2** Review history. · Genome Biology]

Review History

First round of review

Reviewer 1

Were you able to assess all statistics in the manuscript, including the appropriateness of statistical tests used? Yes. Minimal statistics were reported or needed for this work.

Were you able to directly test the methods? No.

Comments to author:

This manuscript describes an approach to benchmarking SVs that is quite different from other methods. It uses a haplotype-resolved assembly as the "truth" to identify the number of true positives and false positives (but not false negatives), including more complex SVs than previous benchmarks, as well as inversions, tandem duplications, and interspersed duplications. Overall, I expect this to be a useful new tool for benchmarking SVs that is nicely complementary to existing methods, especially after addressing some of the points of clarifications below.

1. It would be useful to state in the abstract that this approach measures the number of TPs and FPs (and precision), but not FNs or sensitivity.

2. The authors state "This alleviates the need for a curated gold-standard callset so that any genome with a haplotype-resolved assembly of sufficiently high quality may be included as a benchmark, and can help validate SVs in repetitive regions because explicit breakpoints are not compared." It's probably more accurate that this method complements curated gold-standard callsets rather than "alleviates the need for" them. Consortia like GIAB use community curation of putative FPs and FNs as a way of demonstrating that the benchmark set/benchmarking tool reliably identifies FPs and FNs across a diverse set of variant callers and technologies. While it takes time and tends to result in more conservative (and older) benchmarks, these benchmarks complement efforts like TT-Mars by getting broader curation from the community, and TT-Mars complements other benchmarks by including more challenging variants.

3. I think the methods in the "Validation" section are likely quite important for getting reliable results from TT-Mars, so I think it would be useful at least to include a supplementary diagram describing the process and the various regions that are excluded from the assembly. For example, it's not clear how reads are mapped back to the assembly, whether they are haplotype-partitioned first, or what thresholds are used. I'd expect you get quite different results for older HGSVC assemblies vs the newer trio-hifiasm-based HiFi assemblies, particularly in segmental duplications, which are better resolved by hifiasm, as well as due to better haplotype-partitioning in the newer assemblies. Could the authors include more details about exactly which assemblies the authors used from HGSVC and HG002? Could the authors include more information about what fraction of the genome/assembly is excluded, and how this is influenced by the assembly quality? I could imagine naive users getting inaccurate benchmarking results because they don't understand the limitations of the particular assembly they use as the benchmark.

4. Why are the numbers of TPs, FNs, and NA substantially smaller in Table 2 than Table 1 for TT-Mars using DELLY and pbsv? Maybe variants <50bp are included in Table 1? If so, this would be good to clarify

5. In Fig 3A, do the authors know why there is high variability in the TP rate within pbsv (and to some degree within the other callers) between samples?

6. It's great that TT-mars benchmarks interspersed duplications, so it would be useful to have

more details: When simulating interspersed duplications, were they exact duplicates? How do the authors expect sequence identity might affect benchmarking results? Also, are there any limits in size (e.g., because very large insertions like an extra copy of a large segdup are sometimes not called by assembly-based callers)? What parameters were used with minimap2 when aligning duplications?

7. The authors state "On the ten pbsv callsets, TT-Mars and dipcall+truvari have agreeing classification for 83.8%-86.9% of the analyzed calls (Figure 4b), although TT-Mars analyzes 1,497-2,229 more insertions/deletions than dipcall+truvari... There was no association between length (Figure 3b) nor genomic organization (Supplementary Figure 3) for whether TT-Mars and dipcall+truvari provided a validation." It seems like there was some association with TRs based on manual inspection? Could the authors computationally see how many discrepant calls had multiple large insertions and/or deletions in a TR in the dipcall vcf?

8. I appreciate the inclusion of the manual curation results, including summarizing classifications in the text and supplementary figs, but i could not find information about how the authors classified each figure in the supplementary info 4.1 and 4.2. It would be useful to have these annotated examples to better understand the discrepancies between benchmarking approaches.

9. It's not clear to me whether the improvements in complex variants result from minimap2-dipcall breaking up some SVs into multiple insertions/deletions and Ira not breaking up the SVs into multiple events, or if TT-Mars is able to combine multiple insertions/deletions in a way that truvari can't handle. For example, could you just use Ira with dipcall and truvari and get similar results for benchmarking as TT-mars? It would be useful to more clearly highlight examples of TT-Mars reconciling different representations of variants, and the types of different representations it can reconcile.

10. The authors state "Fortunately, most methods had accuracy within 1-2% across all genomes." In Fig 3A, the TP rate appears to vary by 5-10% across all genomes. Could they clarify where the 1-2% comes from?

11. It would be useful to know the origins of the centromere and tandem repeats files used by the authors, since these can be defined in many ways

12. It appears TT-mars might not output TPs and FPs as vcf files yet, which I expect would be useful

Reviewer 2

Were you able to assess all statistics in the manuscript, including the appropriateness of statistical tests used? Yes. No issues.

Were you able to directly test the methods? No.

Comments to author:

Looks like a useful, practical tool that my lab will certainly be trying out!

Major comments:

- I found the overall method description hard to follow. The first sentence talks about "reads matched to the same genome", but AFAICT reads play no part in the comparison, rather the tool compares an SV set from an SV caller with haplotypes from an assembly aligned to a reference. If this is not correct then I have wholly missed the point.

- The reported increase in assessed variants vs. truvari is questionable, IMO, because we do not know if that increase in variants reflects actually sound calls that can be compared. It would be nice if there were some analysis of the FPs/TPs of these additional variants (i.e. variants not covered by truvari)

- I think the decision to only assess TPs and FPs, and not to attempt to report FNs is really unfortunate. I don't see why you can't (crudely, at least) estimate FNs in a truth assembly if you can compare that assembly to a reference. FNs, particularly, as the authors point out, FNs in medically important genes, are just as important as FPs, IMO.

- There is no attempt, it seems, to assess genotype accuracy, which again is an unfortunate missing feature. Given that there are a number of other tools in the space, it seems like this would be a helpful and distinguishing feature to assess genotypes when they are available in a prediction set. It would be okay, IMO, to only assess genotypes when the truth assembly can be confidently judged to include both haplotypes within a given region.

- I think the tool is limited by the view that individual variants from a variant set can always be meaningfully compared to a truth set. In reality SVs are often, as the authors point, reported as sets of individual edits. While the authors attempt to address this by searching around reported breakpoints and comparing to the haplotypes of the assemblies being compared to, a more interesting approach might be to attempt larger regional comparisons of sets of variants in a region to assess the relative agreement with the assembled haplotypes. Perhaps this is asking too much, but I worry the approach taken here of taking each predicted variant and comparing it to the truth individually will sometimes be misleading.

Minor comments:

There are a number of small typos, nonexhaustively:

- "long semental duplications"

- "limted"

- "have agreeing classification" (odd phrasing)

Dear Dr. Pang,

Thank you for sending us the review and consideration, and thank both reviewers for their detailed comments. We are pleased to read that the referees were favorable of the work, including “Overall, I expect this to be a useful new tool for benchmarking SVs that is nicely complementary to existing methods, especially after addressing some of the points of clarifications below.”, and “Looks like a useful, practical tool that my lab will certainly be trying out!”

We have updated our manuscript to address the issues raised by the reviewers. In summary, these changes include:

1. Analysis of variation of true positive rates between technologies and samples; low validation rates in two PacBio callsets were due to a large number of spurious short variant calls.
2. Addition of two elements of functionality:
 - a. Validating genotype accuracy.
 - b. Providing an estimate of sensitivity.
3. In-depth analysis of differences between validation dipcall+truvari and TT-mars validation approaches on the HG002 callset.
4. Improvements in the clarity of the text.

In particular, none of the issues pointed out by the reviewers required modifications to our original methods beyond the extensions previously mentioned. The only statistic that has changed is a recount of the number of SVs produced by dipcall that likely match SVs in orthogonal alignments, from 47 to 48.

Sincerely,

Jianzhi Yang,

Mark Chaisson

Reviewer #1:

This manuscript describes an approach to benchmarking SVs that is quite different from other methods. It uses a haplotype-resolved assembly as the "truth" to identify the number of true positives and false positives (but not false negatives), including more complex SVs than previous benchmarks, as well as inversions, tandem duplications, and interspersed duplications. Overall, I expect this to be a useful new tool for benchmarking SVs that is nicely complementary to existing methods, especially after addressing some of the points of clarifications below.

We thank the reviewer for their time to review the manuscript and their comments. We have included a response to each point below.

1. It would be useful to state in the abstract that this approach measures the number of TPs and FPs (and precision), but not FNs or sensitivity.

A similar comment was made by reviewer 2 (comment 3). We first emphasize that the comparison against assembly content as opposed to call content is an abstraction that does not support measuring FN.

We have significantly restructured the abstract to conform to the 100-word limit for the methods paper, as suggested by the editors. To account for the comment, part of the rephrased abstract states:

Here we provide an alternative approach, TT-Mars, that takes advantage of the recent production of high-quality haplotype-resolved genome assemblies by providing false discovery rates for variant calls based on how well their call reflects the content of the assembly, rather than comparing calls themselves.

We feel that by explicitly stating that TT-Mars uses assemblies to assess the false-discovery rate for methods emphasizes the novel approach of comparing assembly-based content rather than call based content. Furthermore, to address the comment made by reviewer 2, we have added an additional module to assess false negatives.

We have added the following text describing the reporting of false negatives to the text:

The approach that compares genome content rather than callsets inherently does not support measurements of missed calls, or false negatives (FN). To enable reporting FN, TT-Mars provides the option to report false negatives (FN) by comparing against a callset offered as ground truth.

Variants at least 50 bases produced by dipcall on autosomal and X chromosomes are used as the truth set for annotating false negatives. Calls overlapping with contigs that are trimmed out (Figure 1a) are ignored since they may produce duplicated truth calls. Because variants from the candidate callset are not validated by matching calls, dipcall variants within a specified number of bases of validated candidate calls (default 1kb) are considered matched. Remaining unmatched calls are reported as FN.

We report the following recall results:

The average recall of Wham, LUMPY, DELLY and pbsv are 7.7%, 13.1%, 16.0% and 88.7% respectively.

2. The authors state "This alleviates the need for a curated gold-standard callset so that any genome with a haplotype-resolved assembly of sufficiently high quality may be included as a benchmark, and can help validate SVs in repetitive regions because

explicit breakpoints are not compared." It's probably more accurate that this method complements curated gold-standard callsets rather than "alleviates the need for" them. Consortia like GIAB use community curation of putative FPs and FNs as a way of demonstrating that the benchmark set/benchmarking tool reliably identifies FPs and FNs across a diverse set of variant callers and technologies. While it takes time and tends to result in more conservative (and older) benchmarks, these benchmarks complement efforts like TT-Mars by getting broader curation from the community, and TT-Mars complements other benchmarks by including more challenging variants.

We have changed the text:

This alleviates the need for a curated gold-standard callset so that any genome with a haplotye-resolved assembly of sufficiently high quality may be included as a benchmark,

To the following:

This complements the validation method by the curated gold-standard callset from the GIAB such that any genomes with a haplotye-resolved assembly of sufficiently high quality may be included as a bench-mark,

We note we have currently used this approach to complement Truvai for annotating TOPMed SV calls, exactly as the reviewer states.

3. I think the methods in the "Validation" section are likely quite important for getting reliable results from TT-Mars, so I think it would be useful at least to include a supplementary diagram describing the process and the various regions that are excluded from the assembly. For example, it's not clear how reads are mapped back to the assembly, whether they are haplotype-partitioned first, or what thresholds are used. I'd expect you get quite different results for older HGSVC assemblies vs the newer trio-hifiasm-based HiFi assemblies, particularly in segmental duplications, which are better resolved by hifiasm, as well as due to better haplotype-partitioning in the newer assemblies. Could the authors include more details about exactly which assemblies the authors used from HGSVC and HG002? Could the authors include more information about what fraction of the genome/assembly is excluded, and how this is influenced by the assembly quality? I could imagine naive users getting inaccurate benchmarking results because they don't understand the limitations of the particular assembly they use as the benchmark.

We address the multiple issues raised in this comment sequentially below:

- a. Include a supplementary diagram describing the process and the various regions that are excluded from the assembly.

We have modified Figure 1 to demonstrate which regions from the assemblies are included for validation. These are regions that align to the reference (not missed alignment). We highlight the

removal of overlapped portions of contigs so that a 1-1 orthology map may be created. The regions missing from the 1-1 orthology map are excluded from validation.

- b. For example, it's not clear how reads are mapped back to the assembly, whether they are haplotype-partitioned first, or what thresholds are used.

Our original text about our assembly filter required additional details. We used an assembly score to assess the quality of assemblies by mapping reads to the assembly (both haplotypes), and checking for contiguity of reads across the genome, and flagging bins that are not contiguously spanned by reads as unreliable. This approach only requires a minimal number of reads to map continuously so that heterozygous structural variants are not flagged as discontinuous. Furthermore our filter considers regions that are potentially misassembled rather than mapping artifact, so the regions that we consider low quality are short, and flanked by high-coverage regions.

We have added the following text to make our description of the assembly quality filter more clear:

Alignments of reads back to assemblies are used to flag potential misassemblies that should not be used to validate calls. Correctly assembled short intervals (e.g. 100 base windows) should have reads that map contiguously from at least one haplotype. We define an interval as correctly assembled if at least five reads are aligned starting/ending at least 1kb bases before/after the interval. And misassembled if this condition does not hold, and the intervals in the flanking 1-3kb are supported. These parameters are defined by the data to stringently detect misjoins in assemblies and not be affected by read mapping artifacts in repetitive DNA.

Finally, we note the assemblies are high quality. We detected five misjoins in each haplotype of HG002, and no callset had variants that overlapped these regions.

- c. I'd expect you get quite different results for older HGSVC assemblies vs the newer trio-hifiasm-based HiFi assemblies, particularly in segmental duplications, which are better resolved by hifiasm, as well as due to better haplotype-partitioning in the newer assemblies.

We agree. In this manuscript we show a higher rate of validation for pbsv calls using genomes assembled by HiFi reads, however these are against callsets generated from HiFi reads. Importantly, for other datasets (Illumina based calls), there is less effect from the type of assembly. We have modified Figure 3 to facet points based on the data type used to construct the assembly. Moving forward we expect all assemblies used for TT-Mars to be from HiFi or

equivalently accurate methods.

d. which assemblies the authors used from HG002 and HG002

HG002: The HG002 assembly we used can be found here:

<https://zenodo.org/record/4393631#.YY9NA2DMKbg>, these two files:

NA24385.HiFi.hifiasm-0.12.hap1.fa.gz, NA24385.HiFi.hifiasm-0.12.hap2.fa.gz.

HGSVC assembly: http://ftp.1000genomes.ebi.ac.uk/vol1/ftp/data_collections/HGSVC2/.

Sample genome HG01505, HG01596, HG00096, HG03009, HG00171, HG00864, HG01114 are CLR assemblies. Sample genome HG00513, HG00731, HG00732 are HiFi assemblies.

e. Could the authors include more information about what fraction of the genome/assembly is excluded, and how this is influenced by the assembly quality?

We have added

Supplementary Table 1 includes the fraction of genome excluded by TT-Mars. This approach excludes 254-289 Mbp of sequence in the reference, and an average of 34 Mbp of overlapping sequences from the 10 assembly.

4. Why are the numbers of TPs, FNs, and NA substantially smaller in Table 2 than Table 1 for TT-Mars using DELLY and pbsv? Maybe variants <50bp are included in Table 1? If so, this would be good to clarify

Our Table 1 includes all calls produced by each method, including all variants above 10 bases. While Table 2 includes calls above 50 bases. We have updated the Table 1 and 2 captions:

We have updated the captions of Table 1 and Table 2 to be more specific. The table 1 caption has changed from:

TT-Mars results on callsets from four SV discovery algorithms on HG002.

to:

TT-Mars results on callsets from four SV discovery algorithms on HG002, including all calls larger than 10bp produced by each method.

Table 2 has changed from:

Comparison of TT-Mars and GIAB benchmarks on HG002. The two methods have the same classification results on most analyzed calls, while TT-Mars analyzed more candidate calls on all the four callsets.

to:

Comparison of TT-Mars and GIAB benchmarks on HG002. The two methods have the same classification results on more than 96% of the analyzed calls, while TT-Mars analyzed more candidate calls on all the four callsets. A length filter (50bp - 10Mbp) is applied and only deletions and insertions are included to match the truvari parameter settings.

5. In Fig 3A, do the authors know why there is high variability in the TP rate within pbsv (and to some degree within the other callers) between samples?

We considered data type (HiFi or CLR), read depth, and fraction of the reference mapped by the assembly, and variant size as potential variables that affect TP rate.

We found that the variability in TP rate was primarily due to the difference between CLR and HiFi callset quality, with an exacerbating factor of two CLR callsets (HG01596 and HG01114) that had a particularly high number of short indel calls (< 20 bases) and a low TP rate. When

we consider calls that are at least 30 bases, the TP rate is within 91.7-94.8%, including these genomes.

We have highlighted this in the text, changing the sentences:

The TP rates on short-read callsets range between 63.9%-85.2%, while pbsv calls have TP rates from 71.4% to 94.3%.

and

Among the TP calls, genotype matched rates of DELLY are from 90.1% to 92.1% and of pbsv are from 73.9% to 90.5%.

to:

The TP rates on short-read callsets range between 63.9-85.2%, while pbsv calls have TP rates from 71.4% to 94.3% considering all calls, and 91.7-94.8% for calls ≥ 30 bases.

and:

Among the TP calls, genotype matched rates of DELLY are from 90.1% to 92.1% and of pbsv are from 73.9-90.5% (78.3-91.2 for $SV \geq 30$).

We furthermore include supplementary analysis of true positive rates versus assembly mapping and a comparison of validation counts by length, reproduced below.

The short read callsets are dominated by an outlier sample, HG00513, that has a considerably larger BAM file than the other callsets. This could either cause problems in SV discovery, or need more parameter optimization. We have added the comment to the manuscript:

The sample HG00513 has lower TP rates among all short read SV discovery algorithms, but also has roughly double the input size of other samples that may have required parameter optimization.

We have also included analysis of input sample size and TP rate to the supplementary materials.

Reasons for variability of TP rate within pbsv calls

- a. CCS samples have higher analyzed rate, covered bases and TP rate compared to CLR samples, covered bases and TP rate have positive correlation.

Supplementary Figure 3. The true positive rate versus fraction of the genome covered by contig alignments. The total genome size includes centromeres but no alternative haplotype sequences. The three points in the upper right corner correspond to HiFi assemblies that have both higher quality, as well as higher quality callsets.

Supplementary Figure 3. True positive and total counts by size for short (< 50 base) calls. HG01596 and HG01114 callsets have fewer validated calls, particularly short (< 20 base) calls.

b. HG000513 all short read callers having more calls.

Supplementary Figure 4. TP rate among short read algorithms, along with BAM size. The HG00513 BAM is roughly twice the size of all other bams, and may have had influence on the lower callset accuracy.

d

Supplementary Figure 5. The number of analyzed calls per genome. This highlights the larger number of calls made in the HG00513 genome.

6. It's great that TT-mars benchmarks interspersed duplications, so it would be useful to have more details:

a. When simulating interspersed duplications, were they exact duplicates?

The duplications are simulated with no divergence. We have modified the sentence:

We ran 1,000 simulations of interspersed duplications with length uniformly taken from 100 to 100,000 on the HG002 assembly.

to:

We ran 1,000 simulations of interspersed duplications with an exact sequence copy and length uniformly taken from 100 to 100,000 on the HG002 assembly.

b. How do the authors expect sequence identity might affect benchmarking results?

Because the alignment method used to detect interspersed duplications can detect divergent sequences (e.g. between species, or low quality reads), we do not expect duplication divergence to be a barrier for detection. Furthermore, clinically relevant duplications will be de novo, or recent, and will be of high identity.

c. Also, are there any limits in size (e.g., because very large insertions like an extra copy of a large segdup are sometimes not called by assembly-based callers)?

Our simulations are up to 100,000 bases, with no effect on validation rate by size. However we note as duplication are larger they are more likely to arise in highly repetitive regions of genomes that may not be assembled.

The TT-Mars approach cannot validate duplications missing from assemblies. We have added this as a point in the discussion:

One caveat of this analysis is duplications are enriched in repetitive sequences [Sharp 2005], and some validations will require high quality assemblies using ultra-long and accurate (HiFi) sequencing technologies.

d. What parameters were used with minimap2 when aligning duplications?

We use Mappy (Minimap2 Python Binding <https://github.com/lh3/minimap2/tree/master/python>) to do the alignment. All the parameters are set as default.

7. The authors state "On the ten pbsv callsets, TT-Mars and dipcall+truvari have agreeing classification for 83.8%-86.9% of the analyzed calls (Figure 4b), although TT-Mars analyzes 1,497-2,229 more insertions/deletions than dipcall+truvari... There was no association between length (Figure 3b) nor genomic organization (Supplementary Figure 3) for whether TT-Mars and dipcall+truvari provided a validation." It seems like there was some association with TRs based on manual inspection? Could the authors computationally see how many discrepant calls had multiple large insertions and/or deletions in a TR in the dipcall vcf?

This is a keen point. There is considerable overlap effect of tandem repeats on disagreeing calls. For pbsv callset on HG00096, the portion of calls two methods disagree that overlap tandem repeats is 92.8% and the portion of calls two methods agree that overlap tandem repeats is 69.3%.

We have updated the manuscript from:

There was no association between length (Figure 3b) nor genomic organization (Supplementary Figure 3) for whether TT-Mars and dipcall+truvari provided a validation, <sentence continued>

To:

There was no association between length (Figure 3b) nor genomic organization (Supplementary Figure 3) for whether TT-Mars and dipcall+truvari provided a validation, however there is considerable enrichment in disagreement between methods in tandem duplications. For example, in HG00096, the 92.8% of calls with disagreeing validation overlap tandem repeats.

8. I appreciate the inclusion of the manual curation results, including summarizing classifications in the text and supplementary figs, but I could not find information about how the authors classified each figure in the supplementary info 4.1 and 4.2. It would be useful to have these annotated examples to better understand the discrepancies between benchmarking approaches.

We have added descriptions for each of the IGV shots for which category they belong in. We have added the text in the supplementary material to clarify this (below). Furthermore, we highlight our comment in the text describing how fragmented calls may be validated in the assembly-based approach: "In 27 cases, a call made by pbsv is fragmented into multiple calls by dipcall, highlighting how the assembly-based approach can account for validation in repetitive DNA where breakpoints are uncertain."

In supplementary materials 7.1: TT-Mars TP, dipcall+truvari FP,

Category I: a call made by pbsv is fragmented into multiple calls by dipcall.

Category II: there are correct variants near by and the validation depends on the genomic region considered by truvari.

Category III: No matched SV in the alignments generated by minimap2 that are used for validation, but do show matched SV(s) in the LRA based alignments.

In supplementary materials 7.2: TT-Mars FP, dipcall+truvari TP,

Category I: no matched SV in the assembly alignment used by dipcall.

Category II: exist matched SVs in the assembly used by TT-Mars.

Category III: other (not category I and not category II).

9. It's not clear to me whether the improvements in complex variants result from minimap2-dipcall breaking up some SVs into multiple insertions/deletions and Ira not breaking up the SVs into multiple events, or if TT-Mars is able to combine multiple insertions/deletions in a way that truvari can't handle.

Ira was used because at the time of development it aligned across large events and made the lift over more feasible. However, because no SVs are called using Ira, it does not affect the comparison between TT-mars and minimap2-dipcall+truvari. The improvements in complex variants are largely due to fragmented calls in minimap2-dipcall, which is pointed out when discussing differences between validation approaches.

For example, could you just use Ira with dipcall and truvari and get similar results for benchmarking as TT-mars? It would be useful to more clearly highlight examples of TT-Mars reconciling different representations of variants, and the types of different representations it can reconcile.

While many SV calls that are fragmented by minimap2 are not fragmented by LRA, it is not a complete solution, and the call-free approach by TT-mars is meant to reconcile differences between alignment parameters that influence either the original calls or truth set.

10. The authors state "Fortunately, most methods had accuracy within 1-2% across all genomes." In Fig 3A, the TP rate appears to vary by 5-10% across all genomes. Could they clarify where the 1-2% comes from?

Our manuscript stated accuracy when we intended to state analyzed rate 1-2% is the analyzed rate difference range. TP rate does vary by 5-10%.

We have modified the text to correct this from:

Fortunately, most methods had accuracy within 1%-2% across all genomes, indicating results are extensible across genomes and, for example, the results in current large-scale short-read based genomics studies likely hold valid.

To:

Fortunately, most methods had accuracy within 5%-10% across all genomes, indicating results are extensible across genomes and, for example, the results in current large-scale short-read based genomics studies likely hold valid.

11. It would be useful to know the origins of the centromere and tandem repeats files used by the authors, since these can be defined in many ways

We have added the following to the data availability:

Centromere and tandem repeat annotations were downloaded from the UCSC Genome Browser for GRCh38 from the cytoBandIdeo and simpleRepeats tables.

12. It appears TT-mars might not output TPs and FPs as vcf files yet, which I expect would be useful

We have added vcf output option (-v / --vcf_out), which will allow TT-Mars to output results as separate vcf files: ttmars_fp.vcf, ttmars_tp.vcf and ttmars_na.vcf.

Reviewer #2:

Looks like a useful, practical tool that my lab will certainly be trying out!

Major comments:

1. I found the overall method description hard to follow. The first sentence talks about "reads matched to the same genome", but AFAICT reads play no part in the comparison, rather the tool compares an SV set from an SV caller with haplotypes from an assembly aligned to a reference. If this is not correct then I have wholly missed the point.

We agree the leading description was unclear. In particular, the reads are used as a quality control on the assembly to make sure the validator is valid, but it conflates the description of what TT-Mars does.

We have removed " with reads matched to the same genome as the assembly " from the description to make the introduction:

TT-Mars uses a haplotype-resolved long-read assembly, and a lift-over map to the human reference to assess each candidate SV call (Figure 2a-c).

2. The reported increase in assessed variants vs. truvari is questionable, IMO, because we do not know if that increase in variants reflects actually sound calls that can be compared. It would be nice if there were some analysis of the FPs/TPs of these additional variants (i.e. variants not covered by truvari)

We added additional analysis on pbsv HG002 callset, which is the HG002 callset that TT-Mars has the most additional analyzed calls than truvari with GIAB benchmark (10,966 more calls on the pbsv callset, while TT-Mars has less than 1,476 additional calls on the other 3 HG002 callsets). We plotted length distribution and scatter plot for calls analyzed by both methods and additional calls analyzed by TT-Mars. The length distribution indicates the 2 groups of calls have a similar distribution, although TT-Mars additionally analyzed calls include more small SVs. The scatter plots of TT-Mars results (relative length VS relative score) of the 2 groups of calls demonstrate that TT-Mars have the similar score distributions, the only difference is that the plot of TT-Mars additionally analyzed group is more dense because it contains more calls.

Moreover, we used VaPoR to complement the analysis on the pbsv HG00096 callset. Among TT-Mars additionally analyzed (compared to the dipcall+truvari method) and TP calls, VaPoR has a TP rate (number of TP/number of TP + number of FP) of 89.8%. Among TT-Mars additionally analyzed and FP calls, VaPoR has a TP rate of 67.9%. To compare, among all the TT-Mars TP calls, VaPoR has a TP rate of 91.6%. Among all the TT-Mars FP calls, VaPoR has a TP rate of 68.0%. This shows that VaPoR has almost the same results on the group of additional calls by TT-Mars and all the calls analyzed by TT-Mars, which again indicates that TT-Mars is likely to have consistent results on the entire callset.

3. I think the decision to only assess TPs and FPs, and not to attempt to report FNs is really unfortunate. I don't see why you can't (crudely, at least) estimate FNs in a truth assembly if you can compare that assembly to a reference. FNs, particularly, as the authors point out, FNs in medically important genes, are just as important as FPs, IMO.

We agree that false-negatives are important. In order to address this we have added a module to produce a callset using dipcall and report calls from this filtered set that are distant from TT-validated TP calls as FN.

We have added the following text to the methods:

The approach that compares genome content rather than callsets inherently does not support measurements of missed calls, or false negatives (FN). To enable reporting FN, TT-Mars provides the option by comparing against a callset offered as ground truth. Variants at least 50 bases produced by dipcall on autosomal and X chromosomes are used as the truth set for annotating false negatives. Calls overlapping with contigs that are trimmed out (Figure 1a) are ignored since they may produce duplicated truth calls. Because variants from the candidate callset are not validated by matching calls, dipcall variants within a specified number of bases of validated candidate calls (default 1kb) are considered matched. Remaining unmatched calls are reported as FN.

We have added the following text to the results:

TT-Mars also reports the following recall results: The average recall of Wham, LUMPY, DELLY and pbsv are 7.7%, 13.1%, 16.0% and 88.7% respectively.

Finally, we have modified the discussion to state that we provide FN estimates.

4. There is no attempt, it seems, to assess genotype accuracy, which again is an unfortunate missing feature. Given that there are a number of other tools in the space, it seems like this would be a helpful and distinguishing feature to assess genotypes when they are available in a prediction set. It would be okay, IMO, to only assess genotypes when the truth assembly can be confidently judged to include both haplotypes within a given region.

We have added genotype assessment option (-g / --gt_vali). If the input vcf file has a proper GT INFO field and -g flag is used, TT-Mars will add a column to the output file for the genotype assessment results. If a call is validated (True in the 7th column), the 8th column will be the genotype validation result (True/False). If a call is not validated (False

in the 7th column), the 8th column will be False. If -g flag is used, an INFO field "GT_vali" will also be added to the TP output vcf file (ttmars_tp.vcf) if -v or --vcf_out is set.

The genotype accuracy results have been included in Figure 3A. Furthermore, we have added the following text to describe these results:

DELLY and pbsv output genotype.

Among the TP calls, genotype matched rates of DELLY are from 90.1% to 92.1% and of pbsv are from 73.9-90.5% (78.3-91.2% for SV >= 30).

5. I think the tool is limited by the view that individual variants from a variant set can always be meaningfully compared to a truth set. In reality SVs are often, as the authors point, reported as sets of individual edits. While the authors attempt to address this by searching around reported breakpoints and comparing to the haplotypes of the assemblies being compared to, a more interesting approach might be to attempt larger regional comparisons of sets of variants in a region to assess the relative agreement with the assembled haplotypes. Perhaps this is asking too much, but I worry the approach taken here of taking each predicted variant and comparing it to the truth individually will sometimes be misleading.

This is an insightful comment. While developing TT-Mars, we initially attempted to search combinations of nearby variants to see if two or more variants could be combined in such a way that potentially fragmented calls could be validated. Our results did not show any improvement in validation rates.

6. **Minor comments:**

There are a number of small typos, nonexhaustively:

- "long semental duplications"
- "limted"
- "have agreeing classification" (odd phrasing)

Second round of review

Reviewer 1

The authors have made a lot of improvements in response to the initial reviews, and have done an excellent job responding to each comment, so I have no further suggestions.

Reviewer 2

The reviewers have adequately addressed all my concerns and the paper in general reads very well now. Great job!